# Andrographolide Relieves Post-Operative Wound Pain but Affects Local Angiogenesis

**DOI:** 10.3390/ph15121586

**Published:** 2022-12-19

**Authors:** Yi-Lo Lin, Jiunn-Wang Liao, Shunching Wang, Badrinathan Sridharan, Hsin-Ju Lee, Ai Li, Kai-Ming Chang, Ching-Yang Wu, Siendong Huang, Kai-Ting Chang, Dinesh Chandra Agrawal, Ching-Jung Chen, Meng-Jen Lee

**Affiliations:** 1Graduate Institute of Veterinary Pathobiology, National Chung Hsing University, 250 Kuo-Kuang Road, Taichung 40227, Taiwan; 2Department of Applied Chemistry, Chaoyang University of Technology, 168 Jifeng East Road, Taichung 41349, Taiwan; 3Department of Moleculer Medicine, Koo Foundation Sun Yat-Sen Cancer Center, Taipei 112019, Taiwan; 4Department of Thoracic Surgery, Chang Gung Memorial Hospital, Chang Gung University, Taoyuan 33302, Taiwan; 5Department of Applied Mathematics, National Dong Hwa University, No. 1, Sec. 2, Da Hsueh Rd., Shoufeng, Hualien 97401, Taiwan; 6Department of Basic Research, Holy Stone Healthcare Co., Ltd., Taipei 11493, Taiwan; 7Neural Regeneration Laboratory, Department of Neurosurgery, Neurological Institute, Taipei Veterans General Hospital, Taipei 11217, Taiwan

**Keywords:** andrographolide, *Andrographis paniculata*, post-operative wound, p-ERK, angiogenesis

## Abstract

Andrographolide (Andro), the major constituent of *Andrographis paniculata Nees (Acanthaceae)*, is was known to reduces inflammatory reaction. In the current study, the ability of Andro to reduce pain sensation in a rat post-operative wound model was explored. The hind paws of 18 Sprague-Dawley rats (SD) bearing post-operative wounds received the following three treatments: Saline, Andro via direct injection into the paw (Andro-injected) and Tablet containing Andro + poly (lactic-co-glycolic acid) (PLGA) (Andro-tablet). Von Frey tests assessed mechanical allodynia at 1, 3, 5 h and 1-, 2-, 3-, 4-, and 5-days post-operation. Behavioral analyses were performed to measure reaction threshold and reaction frequencies. Immunoreactivity of p-ERK and GluR1 was examined in the dorsal horn of the spinal cord. Histopathological and immunostaining studies were conducted on paw epidermis to observe the gross morphology and angiogenesis. The threshold for inducing allodynia increased and the reaction frequency reduced in the Andro-injected group compared to the saline-group, at 3 h post-surgery and the effect lasted between 3–4 days. The threshold for inducing pain and reaction frequency for the Andro-tablet group did not differ from the saline-treated group. The levels of p-ERK and GluR1 in the dorsal horn were reduced after Andro treatment. No significant difference in wound healing index was observed between saline and Andro-injected groups, but CD-31 staining showed less angiogenesis in the Andro-injected group. Andro significantly reduced mechanical allodynia compared to saline treatment, both in shorter and longer time frames. Furthermore, Andro influenced the expression of p-ERK and GluR1 in the dorsal horn, and the angiogenesis process in the wound healing area.

## 1. Introduction

Andrographolide (Andro) is found in large quantities in *Andrographis paniculata Nees* (*Acanthaceae*) [1]. *A. paniculata* is used in traditional medicines in China and India as an antidote for snake bites, insect bites, treatment of dyspepsia, influenza, dysentery, malaria, and respiratory infections, and as an antipyretic, detoxicant, anti-inflammatory, febrifugal, and antiphlogistic agent. *A. paniculata* has an analgesic effect against acute infections of the gastrointestinal tract, respiratory organs, and urinary system. This plant species is rich in 20-oxygenated flavonoids and labdane diterpenoids [1]. Andro exhibits many biological effects such as as immune regulation, hepatic protection, cardiovascular protection, antiviral, anticancer, and antidiabetic [2]. Andro reduces the lipopolysaccharide (LPS)-induced pro-inflammatory factors in microglia [3]. Previously, we reported that Andro suppressed TNF-induced astrocytic IL-1, IL-6, and TNF in astrocytes in vitro [4] and that Andro reduced allodynia in a spared nerve injury model [5].

An operation usually involves cutting of the skin leading to the damage of neighboring blood vessels. This poses additional concerns regarding the repair, cure, and control of pain in body locations besides the target organ or tissue to be operated. The skin is far more sensitive to pain than the targeted organ or tissue, and the allodynia generated is not to be neglected. For endoscopic surgery, local bupivacaine or lidocaine injection around the post-operative wound area provides pain relief for several hours; however, the post-operative pain typically persists up to several days and therefore emerges as an important concern for the development of pain-relieving drugs for this wound type.

In this study, we examined whether Andro could be used for relieving pain in the skin of a post-operation wound. We tested the injection of Andro dissolved in saline, as well as tablets containing Andro mixed with PLGA, in a post-operative wound allodynia model [6]. Several reports have shown that the non-NMDA/a-amino-3-hydroxy-5-methyl-4-isoxazole propionate (AMPA) receptor is involved in the spinal transmission of pain behavior after incision [7]. This AMPA is regulated by protein kinase C (PKC), and further signaling to the nucleus occurs via PKA, MEK, ERK in the neurons and microglia [7]. Apart from the behavior test, we wanted to validate the reduction in allodynia in Andro-injected rats through downregulation of phosphorylated extracellular signal-regulated kinases (pERK-1/2) and GluR1 subunit of the AMPA receptor in the spinal cord tissue. Further, we also investigated whether the wound healing process was altered by the administration of Andro. It is known that NSAIDs inhibit angiogenesis by directly affecting the endothelial cells. The mechanism involves inhibition of mitogen-activated protein (MAP) kinase (ERK2) activity, and ERK nuclear translocation. It has prostaglandin-dependent and prostaglandin-independent regulation systems. Systemic use of ibuprofen is known to be anti-proliferative during wound healing [8]. Since Andro is identified as an inhibitor of NF-KB, a key regulator of pro-inflammatory reactions, and some of its analgesic effects might act through inhibition of inflammatory reactions [4], it can be claimed that andrographolide shares some pathway with NSAIDS. Therefore, it would be interesting to test whether Andro affects the wound healing process. We examined the local wound healing process by grading HE- and trichrome-stained sections, as well as immunoreactivity of laminin and CD31 for basal lamina and angiogenesis.

## 2. Results

### 2.1. Von Frey Tests: Reaction Threshold (Left and Right)

#### 2.1.1. The Reaction Thresholds of the Right Hind Paw in the Injection Group Were Significantly Increased Compared to the Saline Group

The reaction threshold was set at a point where out of 10 tests, over 50% movement of the paw was observed. Immediately after the operation, the reaction threshold of all groups decreased. The threshold of the injection group did not show a significant difference compared to the saline group at 5 h, while the reaction threshold of the injection group increased compared to the saline group 1–5 days P.O. (Figure 1A). There was no difference between the PLGA-tablet group and the saline group at any of the time points tested (Figure 1A).

#### 2.1.2. The Reaction Thresholds of the Left Paw in the Injection Group Were Significantly Increased Compared to Saline Group

After the operation, the thresholds of the left paw in all groups decreased significantly. The reaction threshold of the injection group did not significantly differ from the saline group after 1, 3, 5 h, and one-day P.O. However, the reaction threshold increased compared to the saline group at 2-, 3-, and 5-days P.O. (* in Figure 1B). There was no difference between the PLGA-tablet group and the saline group at any of the time points tested (Figure 1B).

#### 2.1.3. The Reaction Frequency of the Right Paw in the Injection Group Was Significantly Reduced Compared to the Saline Group

To obtain the data for the reaction frequency, data collected when using 2 g and 4 g von Frey hairs were selected, and the respective values were analyzed. Reaction frequency was determined by the following formula: (number of withdrawal reaction)/10) × 100%. A lower frequency rate indicates less mechanical allodynia. When the 2 g von Frey hair was used (Figure 2A), the paws of post-operative rats from the saline-treated group moved about 50% of the time tested indicating an allodynia behavior (Figure 2A, saline R). When treated with Andro by injection (injection group), the reaction frequency decreased by 20% or lower, and was statistically significant compared to the saline-treated group, at 1, 3 h, and 2–5-days P.O. (Figure 2A, *). When treated with PLGA-tablet (Andro tablet-R), the results did not show much difference when compared with the saline group, demonstrating only a marginal increase (Figure 2A).

For the data with 4 g von Frey hair, post-operation, in the saline group, the paw moved about 60% during the time tested, indicating an allodynia behavior (saline-R, Figure 3B). When treated with Andro by injection (Andro-inject-R), the rate reduced by 35% or lower, and was found to be significantly different compared to the saline-treated group, at 2–5 days P.O. (Figure 2B). When treated with PLGA-tablet (Andro-tablet-R), the values showed only a marginal increase when compared to the saline group (Figure 2B).

#### 2.1.4. The Reaction Frequency of the Left Paw in the Injection Group Was Significantly Reduced Compared to the Saline Group

In the left paw, the saline-treated group demonstrated some allodynia behavior when von Frey hair was used. The reaction frequency of the left paw was there but lower than the wounded right paw. This was expected as how the mirror pain would be (compare Figure 3 for mirror pain to Figure 2 for local pain). The reaction frequency was slightly higher at 3-days P.O. to reach about 30% and then reduced at 4–5 days P.O. However, it was not higher compared to the elicited behavior in the right paw and could be characterized as mirror pain, as the values did increase after the operation. When treated with Andro by injection (Andro-inject-L), the reaction frequency reduced to about 0–5 % and was found to be significantly different than the saline-treated group, at 2–5-days post-operation (Figure 3A). Treatment with the combination of Andro and PLGA tablet (Andro-tablet-L) did not show much difference when compared with the saline-treated group but increased gradually throughout the testing period (Figure 3). When data with 4 g von Frey hair were used, the Andro-injected group performed better than the saline group only at 5-days P.O. (Figure 3B).

### 2.2. Phospho-ERK and GluR1 Staining

Behavioral analyses clearly showed that only Andro-injected rats exhibited a reduction in allodynia with a higher reaction threshold than the saline-injected and Andro tablet groups. To validate the observations in behavioral analyses, we checked the expression of p-ERK in spinal cord tissues (Figure 4A–F). Saline-injected rats, at the end of 5-days post-surgery showed prominent expression of p-ERK in the spinal cord neurons of the dorsal horn region, especially at the laminae I–II (marked with an arrow in Figure 4A), indicating the initiation and conduction of post-operative pain (Figure 4A–C). However, Andro injection significantly reduced p-ERK expression, as illustrated in Figure 4D–F, at the dorsal horn region (marked with an arrow in Figure 4D). GluR1 subunit of the AMPA receptor also plays a vital role in surgical pain, and the saline-injected rats showed significant upregulation of GluR1 in the neuropil of laminae I–II of the dorsal horn post-operatively (Figure 5A,B). Meanwhile, the Andro-injected rats showed a significantly reduced expression of GluR1 (Figure 5C,D).

### 2.3. HE-Based Pathological Evaluation

As some pain killers have been reported to delay the progress of wound healing, we continued to test whether andrographolide would also exhibit this effect. At the end of the behavior test (5-days post-operation), the tissue from the wound periphery was removed and pathological examination was performed. As the Andro-tablet group did not test positive for pain alleviation, we only tested tissues from the saline and the Andro-injected groups. The right paw was the site of surgical intervention. The left paw was also collected from the same animal to serve as a non-operated control group. The non-operated paw had less edema and swelling. The operated paws in both the saline and Andro-injected groups demonstrated a similar extent of swelling and edema. The granular tissue within the wound site and the neighboring, recovering epidermis and dermis were analyzed for their pathological characteristics, as shown in Table 1 and summarized in Table 2, with criteria described in material and methods.

In the footpads of the non-operated paws, the epidermis could be recognized as a layer full of hematoxylin-stained nuclei and an overlying pink eosin staining (Figure 6A). The underlying dermis (d) contains collagen fibers, elastin fibers, blood vessels, lymphatic vessels, nerve endings, fibroblasts, and macrophages. The most prominent findings in this study are the collagen-rich blood vessel (Figure 6A, arrow) and eosin-stained ECM fibrous components. Minimal mononuclear cell infiltration was found in the dermis.

In the operated saline-treated group (Table 1, upper, saline, R), the footpad wounds of all six animals showed focal moderate epidermal crust (grade 2–3 except 1), moderate/severe (grade 4) inflammation, newly formed (grade 3) angiogenesis, and moderate granulation layer (grade 2). The re-epithelialization grade was not as uniform among the six animals. The variation was consistent with the date of operation, in which three rats operated on the first date were graded 3, and three rats operated on the second date were graded 2 (Table 1, upper, saline, R). Figure 6B demonstrates a representative wound area from the rat 18R.

In the right footpad of the operated, Andro-injected group, the grades for the six animals were more variable and not as uniform as the saline group (Table 1, lower, Andro-inject, R). The grade for the crust in the epidermis varied between 2 and 3, which is not statistically different from the saline group. For the inflammation reactions in the dermis, two out of six rats showed a lower degree of inflammation (I11R, I21R), which confirms our previous observation that Andro reduces inflammatory cytokines [3,4], although the significance of the difference was not high from the saline group (Table 2). Figure 6C demonstrates a representative wound area from the rat 21R. The crust of the epithelium and re-epithelization in the saline group (18R) is grade 3, grade 3 (Table 1, upper, saline, R, 18), and in Andro-injected group (21R) is grade 2, grade 2 (Table 1, lower, Andro inject, R, 21).

Data gathered in the HE pathological study demonstrated that three out of six rats in Andro-injection group had less angiogenesis (grade 2, 19R, 21R, 23R), compared to the saline group. Figure 7 demonstrates three representative figures of the normal group (A, 20L, grade 3), saline group (B, 20R, grade 3), and Andro-injection group (C, 21R, grade 2). In the non-operated paw (Figure 7A), blood vessels were present in the dermis area underneath the basal lamina. This was stained by hematoxylin in the endothelial cell and eosin in the lamina of the vessel (arrow in Figure 7A). In the operated saline-treated paw, the regenerating dermis from the active granulation site of the forming vessels was visible encircled by cells stained with hematoxylin in the cross sections. (d), (arrow in Figure 7B). However, there were relatively fewer structures in (Figure 7C). As the difference was not prominent from the staging results (Table 2, *p*-value = 0.045 for type 2, = 0.076 for type 3, *t*-test), further testing of the angiogenesis using immunostaining for laminin, which stains basal lamina, and CD31, which stains the blood vessels was performed

### 2.4. Laminin Staining for Pathological Evaluation

The skin tissues of the paws were collected and sectioned in the orientation demonstrated in Figure 1. Similar to the HE photos, the incision site was located on top of the figure, and the blade-sectioning plane aligns with the figure background plane. Figure 8 showed the representative figure of saline vs. Andro-treated wound area. In the area located further away from the wound site (green arrow in A, B), the basement membrane stained highly of laminin and was easily recognized by its wave or spike-like shape. On top of this was a dense DAPI-stained keratinocyte nucleus which resides in the stratum basale and stratum spinosum. On the other hand, the area near the incision site underwent deconstruction and inflammatory reaction, and these distinct structures were no longer apparent (red arrow in A, B). Near the injury site in the saline group (Figure 8C), underneath the basement membrane, there were numerous laminin-positive vessels which presumably were part of the angiogenesis process during wound healing. On the other hand, although many components were immunoreactive for laminin, there was less laminin positivity with distinct vessel structures in the Andro-injected rats (Figure 8D). When 1000 × 1000 pixel frames from the center of the wound were analyzed with image J as described in the methods section, the data suggested a trend of reduction in the level of angiogenesis in the Andro-treated rats (*p*-value 0.09958 in Wilcoxon rank-sum test) (Figure 8E).

### 2.5. CD31 Staining for Blood Vessels

Figure 9 shows the representative figure of saline vs. Andro-treated wound area. The distinct shapes of the epidermis and dermis could be differentiated by the densely packed DAPI nucleus in the basal lamina and were used to locate the rest of the structures. The areas immediately beneath the epidermis and in the center of the wound are granulation tissue, which underwent fibroblast proliferation and regrowth of blood vessels [9]. In the Andro-treated groups, (Figure 9A–C), cross-sections of hollow blood vessels were apparent (arrow in A). Some have branches and may represent vessels that were not dead (arrow in C). In the saline treated group (Figure 9D–F), some clusters of cells showed elevated CD31 immunoreactivity, and some were observed surrounding a hollow area, but these CD31 immunoreactive cells were round in shape and did not further differentiate into a thin layer of endothelial cells that align with an orientation parallel to the vessel axis (for example, arrow in D). CD31-positive blood vessels at the cross-sections were marked and measured for signal intensity and a sum of the intensities measured from different sections of the saline and Andro groups was added. It was observed that CD31 immunoreactivity for the two groups demonstrated a statistically significant difference (Wilcoxon rank-sum test, *p* = 0.0303).

## 3. Discussion

During the operations, skin and muscle incisions via surgical blade were inevitable and presented an additional target for repair and pain control in addition to the target tissue to be operated. Contrary to common thought, Gould and others reported that post-operative pain may be severe even with sufficient parenteral opioid administration [10]. To help understand this type of pain and its therapy, a rat model of post-operative pain was developed. Immediately after the operation, the pain behavior was tested via a reaction threshold elicited by mechanical pressure by von Frey hair. Subsequently, pain behavior was significantly lower than in pre-incision or non-operated rats [6]. The behavior index was used to estimate the level of pain sensation.

Our data in this study demonstrate that Andro when administered as saline solution, reduced the reaction threshold by 3-h post-operation. Tablets containing Andro blended with PLGA powder did not reduce the threshold level at any time point tested. The experiment using PLGA-embedded Andro was to test for a prolonged analgesic effect with supposedly slow release. However, the effect of Andro administered with saline was delayed compared with other drugs such as morphine, and the PLGA-embedded Andro did not elicit its effect before the time period where allodynia ceased to differ with the non-treated group. Therefore, we did not further discuss the usage of Andro-embedded in PLGA.

Our data demonstrate that Andro did not significantly reduce the reaction threshold in the von Frey test until 3 h post-operation. When the reaction frequency was examined with the hair weight fixed at 2 g, the value was significantly lower than the saline-treated group 1 h P.O. These data suggest that immediately following operation, the effect of Andro starts to take effect, and becomes more profound at 2-days P.O.

### 3.1. Reduced Allodynia Due to pERK Downregulation in Andro-Injected Rats

Activation of spinal ERK1/2 by phosphorylation was reported in mechanical allodynia in a rat model of post-operative pain, while MKP-3 and ERK1/2 specifically were reported to be crucial for the resolution of incisional pain [11,12,13]. Another member of the MAPK family, the p38, was reported in neuropathic pain and also in other types of pain sensation [14,15]. We previously reported that the phosphorylated p38 was increased after incision, and reduced following Andro injection locally [16]. Phosphorylation of ERK1/2, which was crucial in resolution of the incisional pain, was reported in this paper (Figure 4). MKP-3, an ERK2-specific phosphatase [17], was a possible target for the drug to alleviate incisional pain. However, as we observed both reduced phosphorylation of ERK1/2 and p38, it is possible that Andro was acting on a target further up-stream to both the ERK 1/2 and p38 rather than specifically to MKP-3. As Andro was reported to be a NF-KB blocker, this signaling could be downstream to the possible NF-KB inactivation.

### 3.2. Reduced Allodynia Due to GluR1 Downregulation in Andro-Injected Rats

Apart from the p-ERK-mediated acute post-operative pain, AMPA receptors (non-NMDA) play an important role towards initiation and persistence of surgical incision pain. During incisional pain, dorsal horn neurons are sensitized leading to hyper-phosphorylation in GluR1 subunit of AMPA receptor by PKC. The p-GluR1-containing AMPA receptors are trafficked to the cell surface, where they aid in the influx of calcium for conduction of nociceptive stimuli [18,19]. GluR1 is considered as a potential therapeutic target for management of post-operative pain and in our study, Andro showed a significant reduction in GluR1 expression in the dorsal horn region of the spinal neurons compared to the saline-injected rats where we found an upregulated expression of GluR1. Although the GluR1 could be induced acutely after incision in only 3 h [20], in our study, the reduction in the von Frey threshold starts acutely 1 h after operation in part of the assay, and the inhibition is sustained until 5-days post-operation in part of the assay. The immunoreactivity was only tested for 5-days post-sacrifice, and the GluR1 results corroborated the von Frey results.

Several drugs or compounds that target the GluR1 proved to reduce pain or incisional pain. NBQX, a blocker of non-NMDA receptor, eliminated the responses to mechanical stimuli after incision in wide dynamic range (WDR) neurons, and the pinch receptive field (RF) expansion into uninjured areas of the paw was significantly reduced [21]. Stargazin is a transmembrane protein that regulates synaptic targeting of AMPA receptors [22,23]. Selective down-regulation of spinal stargazin via siRNA inhibits the enhanced surface delivery of GluR1 subunit in rat dorsal horn and alleviates post-operative pain [24]. Antagonists to the AMPA receptor (e.g., tezampanel, topiramate, NS1209) alleviate chronic pain but come with some side effects such as dizziness and sedation [25]. Selective blockade of γ8/AMPA receptors with LY3130481 may suppress nociceptive signaling with fewer CNS side effects [26].

We demonstrated that the alleviated incisional pain via Andro administration was accompanied by a sustained reduction in GluR1 (5-days post-operation). Although andro was reported to act on NF-KB and no direct interaction to GluR1 was reported at molecular level, it was to be expected that the GluR1 might be down-stream to the signaling pathway of the anti-inflammation of andro. Again, andro was pretty tolerated orally. Our model was done when the andro was injected. The water solubility of andro was not great but because of its versatility efforts were invested to provide nano-formula of andrographolide to improve the bioavailability [27,28,29,30]

### 3.3. Comparison between Andro and Cox Inhibitors for Incisional Pain: Andro Performs Better

Various analgesics have been tested for their effects on post-operative pain [31]. Some of them are regularly used clinically, which include morphine sulphate, gabapentin, a gamma-aminobutyric acid (GABA)-mimetic compound, and four COX inhibitors which include celecoxib, naproxen, indomethacin, and etoricoxib [32]. Rats in this study were treated with 150 µg Andro in 150 µL of saline. On systemic spread, it is equal to 0.5 mg/kg for a 300 g rat. The ED_50_ for the COX inhibitors was about 7–20 mg/Kg [31]. However, in our study Andro showed COX inhibition less than 1/10 of already reported inhibitors. We did not test the ED_50_ for the Andro, as our focus was aimed at Andro’s effect on neo-angiogenesis. However, the information that Andro could be effective at 1/10 dose of the ED_50_ of COX inhibitors is still interesting, given that the problem of angiogenesis could be managed by other approaches.

Our data demonstrate that the fastest time for Andro to exhibit an effect was 1 h, in some, but not in all the tests. The effect was maximal at 2–3 days, for all 3 analyses (reaction threshold and frequency for 2 g, reaction frequency for 4 g). In a previous publication, the maximal concentration administered for COX inhibitors was 1 day after operation via gastric gavage, and the maximum effect was achieved at 2–3 days [31]. The pharmacodynamics of Andro, therefore, did not differ much from of COX inhibitors in terms of time taken to reach the maximal effect. In brief, although still far from a conclusive inference, our data suggest that for treating acute pain caused by incision or operation, Andro is at par with other COX inhibitors, or possibly better.

### 3.4. Comparison between the Spared Sciatic Nerve Model and Postoperative Pain Model

Andro reduces allodynia in a spared sciatic nerve model, as reported by our group [5]. The spared sciatic nerve model deals with pathological pain, which is characterized by an amplified response to normally innocuous stimuli, that involves positive feedback of the inflammatory reaction loop regulated by the microglia, neurons, and astrocytes [33], which are located mainly in the central nervous system. The postoperative model, on the other hand, did not inflict specific injury to the nervous system. It mimics the incision to the skin and/or muscle by a surgical blade. Nociceptor sensitization at the incision site causes primary hyperalgesia, amplification in the central nervous system may also contribute. Zahn and coworkers reported that the pathological pain model differs from the post-operative pain model as an NMDA relieved pathological pain and failed in the reduction of post-operative pain. For the pathological pain model, several molecules were reported to regulate synaptic transmission, and NMDA and AMPA were among them [34]. Our data demonstrated that Andro acted on the AMPA receptor in the incisional pain model. Our previous observation on the reduction of pathological pain was likely mediated via AMPA as well.

Our data demonstrated that in the non-operated leg, the reaction frequency reduced as early as 3 h after operation (hair weight set at 4 g) (Figure 3B). As the left leg was not operated on, this describes the so-called ‘mirror pain’ and was mediated through the central component. This corroborated the discovery of other researchers, namely that central sensitization set in fairly quickly for post-operational pain [19]. With the drug injected locally in the paw, the suppression of pain behavior coming from mirror pain, which is mediated by the central element, takes place as early as 3 h post-operation.

In the present work, we reported behavior improvement for post-operative pain and possible problems with wound healing. Contrary to common thought, the pain induced after operation/incision may be severe even with sufficient parenteral opioid use [10]. Like the injured sciatic nerve model, the operation/incision pain also elicited allodynia, but it was a much more common unmet medical need, making alleviation of the post-operative pain surprising and interesting. The possible problem with angiogenesis is new and significant, as this information is much needed for evaluating Andro as a potential drug.

### 3.5. Possible Modulation to the Wound Healing Process by Andro

In the latter part of this paper, we demonstrated that the wound healing process was possibly modulated in terms of angiogenesis via Andro administration. At the end of the behavior test (5-days post-operation), the tissue from the wound periphery was removed and pathological examination on HE- and trichrome-stained sections was performed. For the inflammation reactions in the dermis, a couple of Andro-treated rats 2 showed a lower degree of inflammation, which conforms with our previous finding that Andro reduced inflammatory cytokines. When CD31 staining was tested, we found that the Andro-injected group had reduced blood vessels around the wound periphery.

In the course of wound healing, macrophages provide cytokines to stimulate angiogenesis and build a new extracellular matrix for cell ingrowth, and angiogenic capillary sprouts and within a few days form a microvascular network throughout the granulation tissue. These blood vessels carry oxygen and nutrients to maintain cellular metabolism [9]. The reinvasion of the blood vessels was estimated by measuring the CD31-positive blood vessel cross-sectional area in each section. Our results demonstrated that the CD31 reactivity was reduced in the injected rats. As the regrowth of blood vessels is an important step in wound healing, this would possibly result in delayed wound healing. On the other hand, a study reported that macrophage-related inflammation is an important source for pro-angiogenic factor, and robust inflammation and robust angiogenesis during cutaneous wound healing actually results in scar formation, and moderate or refined angiogenesis may hold the key to reduce scar formation [35]. Therefore, the reduced, or possibly delayed angiogenesis in the Andro-injected wound may not be disadvantageous.

## 4. Materials and Methods

### 4.1. Animals

Animals used were non-SPF female Sprague-Dawley (SD) rats (weight 250 g, 8 weeks old) purchased from and bred by biotech company Biolasco based in Yilan County, Taiwan. After the operation, animals were kept in individual cages in ventilated, humidity- and temperature-controlled rooms with a 12 h light/dark cycle. They received food pellets and water ad libitum. The Ethical Committee for Animal Research of National Chung Hsing University approved all experiments.

### 4.2. Surgery

A longitudinal incision of 1 cm was made with a surgical blade No. 11 on the medial plantar surface of the right hind paw. The incision began 0.5 cm distally from the end of the heel to the first set of footpads. The skin, fascia, and plantaris muscle were all elevated using forceps. The epidermis and dermis were separated by scissors in the area directly underneath the wound (horizontal blue arrow in Figure 10B) to allow the Andro-PLGA tablet to be inserted. The same procedure was performed for saline- and injection-treated groups. The skin was closed with 5–0 nylon sutures, and intraperitoneal antibiotics were given. After surgery, the animals were allowed to recover from the anesthesia in their cages. Wounds dehiscence was checked for before each behavioral examination.

### 4.3. Treatment Groups

Experimental groups were as follows: (i) Pain control (Saline) groups: the rats were operated on and treated with 150 µL saline locally in paw after surgery (n = 6). (ii) Andro injected (injection) group: the rats were operated on and treated with Andro (details of doses see following), which was administered on the paw locally (n = 6). (iii) Andro in tablet (tablet) group: the rats were operated on and treated with andro mixed poly(lactic-co-glycolic acid) (PLGA) tablet inserted underneath the wound (n = 6). Six rats were used for each treatment groups, and total of 18 rats were used. The behaviors were tested consecutively for 0–5 days, but the experiments were conducted in 3 batches, each batch containing different treatments of rats. The individual rat was considered the experimental unit within the studies (details see figure legend). For pathological test and immunohistochemical study, rats were sacrificed at 5-days post-operation, and the samples were collected from the paw area for examination. For pathological grade, one manual grading was carried out from representative slice for each rat.

### 4.4. Behaviour Test for Allodynia Thresholds: Von Frey Hair Test

Mechanical allodynia threshold at the lateral plantar surface of the hind paw was assessed before wound operation (as basal pain threshold), and then testing began 1 h after surgery and continued at 3, 5, h and 1-, 2-, 3-, 4-, and 5-days post-surgery. Mechanical sensitivity was determined by measuring the reaction thresholds, and reaction frequency to von Frey hairs (Stoelting, Wood Dale, IL, USA) using the up and down method as described by Chaplan et al., 1994 and frequency method described by Tanga et al., 2005 [36,37,38]. Animals were habituated over for 2–3 consecutive days by recording a series of baseline measurements. Animals were placed in a plastic cage with a wire net floor and were allowed to habituate 10–15 min before commencement of the testing. The filaments were applied in ascending order, 5 times each at an interval of 2–3 s to the plantar surface of the hind paw [39], and the smallest filament eliciting a foot withdrawal reaction was considered as the threshold stimulus. The reaction threshold (force of the von Frey hair to which an animal reacts in over 50% of the presentations) was recorded. For measuring reaction frequency, 2 g hair and 4 g hair were selected. In three sets of 10 stimulations each when using specific hair, the percentage of withdrawal reaction exhibited was noted. 

The rats arrived in batch of 6, and 3 surgeries carried out at a time by YL, and 3 different treatments were allocated and carried out to these 3 rats without any particular randomization method by a non-author assistant. The von Frey test was carried out by SW and HL, and data analysis carried out by SW and ML. The animals will be excluded from the experiments if the basal pain threshold before surgery is less than 4 g. In our experiments none of the rats were excluded. Animals were weighted before surgery and before each von Frey test. If the body weight were less than 20% of the pre-operation weight, the animal were euthanized and data from this rat discarded. During our experiments none of the animals were excluded. For a schematic sequences of treatments please see Figure 11 that follows. 

### 4.5. Drug Preparation and Administration

Lyophilized Andro powder (purchased from Aldrich, cat number 365,645, 500 mg, 98%) was dissolved in dimethyl sulfoxide (DMSO). It was subsequently diluted in normal saline to 1 μg/μL. It was administered via sub-epidermis injection locally at paw near wound at 150 µL volume per animal soon after completion of surgery and before recovery from anesthetics. For the tablet group, 150 μg Andro powder was mixed with 450 μg of poly(lactic-co-glycolic acid) (PLGA) (Sigma SIP2191) and heated to 70 °C in a tablet pressing mold for 2 h for the PLGA to set.

### 4.6. Collection of Tissue Samples

The rats were deeply anesthetized with sevoflurane and sacrificed by bloodletting or intracardiac perfusion with 4% paraformaldehyde. The spinal cord was removed from the vertebra, and L4 to L6 segments were further fixed with 4% paraformaldehyde before moved to PBS. The paw skin was collected from both the operated and non-operated feet of the same rat. The samples were fixed by 4% paraformaldehyde in PBS overnight, followed by several washes with PBS. The paw was cut in half (orientation see Figure 1A), and half of the tissue was cryoprotected with 30% sucrose and cryosectioned at 10 µM for immunofluorescent staining. The remaining tissue went through graded alcohol to dehydrate, followed by embedding in the wax block. These were stained with hematoxylin-eosin (HE) and masons trichrome stain (MT) for pathological study and grading.

### 4.7. Immunohistochemistry

Wax sections were de-waxed in a series of xylene, graded xylene/ethanol mixture, graded ethanol mixture, and finally incubation in PBS. The sections were stained with HE and Masson’s trichrome. Cryosections were mounted on silane-coated slides, air dried, and blocking with 5% bovine serum albumin (BSA) + 0.2% Triton X-100 in PBS if needed. Sections were then left to incubate with primary antibodies overnight at 4 °C, or 4 h at room temperature before overnight incubation at 4 °C. The primary antibody used was rabbit anti-laminin (1:200, L9393, Sigma, St. Louis, MO, USA), rabbit anti-CD31 (1:50, PA5-16301, Thermo Fisher Scientific), rabbit anti-Phospho-ERK1/ERK2 (1:50, 44-680G, Thermo Fisher Scientific, Waltham, MA, USA), and rabbit anti-GluR1 (SD2010) (1:100, MA5-32344, Thermo Fisher Scientific). The secondary antibody for fluorescent photography was CF™ 488 goat Anti-Rabbit IgG (H+L) used (1:400, SAB4600045, Sigma). Primary antibody omission controls were used for all immunostaining protocols to control nonspecific binding. Fluorescent was performed with a Zeiss Axioscope microscope equipped with charge-coupled device (CCD) camera with appropriate filter sets.

### 4.8. Pathological Grading for Wound Area

The grading was performed according to the criteria set in earlier reports [40,41,42]. Briefly, the crust in the epidermis and degree of inflammation of the dermis was evaluated according to a one-to-five grading system, in which 1 = minimal (<1%); 2: slight (1–25%); 3 = moderate (26–50%); 4 = moderate/severe (51–75%); 5 = severe/high (76–100%) [26]. In this system, the higher the grade, the worse the inflammatory reaction. The extent of angiogenesis, granulation, and re-epithelialization was evaluated according to the criteria reported previously [40,41]. The degree of lesions was graded from one to four. The higher the index, the better the recovery.

### 4.9. Quantification of Immunoreactivity (IR) for Blood Vessels

Immunoreactivity was analyzed by using fluorescence intensity of the photography from 6 independent animals of each treatment group. Photographic images were acquired using the same setting for each experiment for the quantitative comparison. For laminin, 1000 × 1000 pixel frames from the center of the wound were analyzed with ImageJ. All pixels within the area to be measured were analyzed with ImageJ software on a 0 (black) to 255 (white) scale. The number of pixels that had a value larger than 40 (for fluorescence intensity) was calculated and used to estimate the Laminin immunoreactivity in the region of interest. For CD31-stained blood vessels, photos were taken from the junction of the epidermis and dermis in the wound area. The cross-section area of the blood vessels that were stained with CD31 was calculated using ImageJ software were CD31 positive blood vessels were manually circled and measured for signal intensity. The summation of the signal intensities obtained from cross-sections of CD31-positive blood vessels was used as an indication of angiogenesis. CD31 reactivity without the shape of the blood vessel was not counted.

### 4.10. Statistical Analysis

In the statistical analyses for behavior test (Figure 1, Figure 2 and Figure 3), the following methods were used. To avoid the normality assumption, the Kruskal–Wallis rank-sum test (non-parametric one-way ANOVA) and Dunn’s test (non-parametric post hoc test) were used. Calculation was done using R version 4.1.2 (www.r-project.org, accessed on 1 November 2021). P-values were corrected using the Hochberg’s method [43]. For immunostaining of laminin and CD31 (Figure 8 and Figure 9), there was only 2 groups to be compared, and Wilcoxon rank-sum test was used. A test was rejected when *p*-vales was greater than 0.05. The results were showed as mean ± standard deviation. For statistical analyses of the angiogenesis for the HE pathological grading, the degree of lesions was graded from one to four [24] and listed in Table 1. The results were expressed as mean ± standard deviation. The data were analyzed by unpaired Student *t*-test. The level for statistical significance was set at *p* < 0.05.

## 5. Conclusions

To summarize, local injection of Andro into paws of rats reduced allodynia caused by post-operative pain which coincide with the suppression of p-ERK and GluR levels in the dorsal horn. The dose and route of Andro supplementation affected angiogenesis around the wound periphery.

## Figures and Tables

**Figure 1 pharmaceuticals-15-01586-f001:**
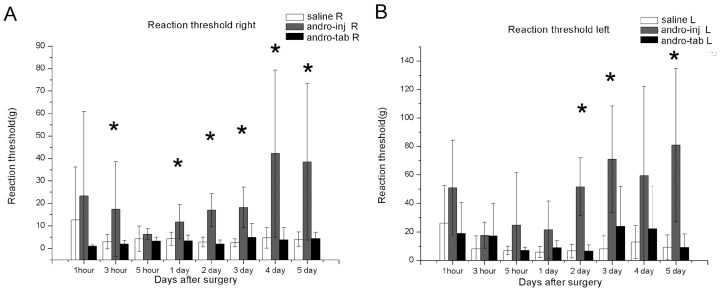
Reaction threshold, an indication of pain sensitivity, was tested for wounds under different treatment conditions. (**A**) Ipsilateral paw (right), (**B**) contralateral paw (left). The mechanical allodynia threshold was assessed by von Frey test. The smallest filament eliciting a foot withdrawal response was considered the threshold stimulus. The data of withdrawal threshold was collected when out of 10 tests, paw movement occurred over 50% of the time. The hind paw reaction threshold was averaged within groups. Saline: rats that had the skin of their right paw cut and sutured, and injected with saline. Andro-inj: rats that had the skin of their right paw cut and sutured, and injected with Andro. Andro-tab: rats that had the skin of their right paw cut, Andro-containing tablets inserted underneath the skin, and sutured. The reaction threshold of the Andro-inj group was significantly decreased compared to saline group (*), while the Andro-tab group was not significantly different to the saline group. This happened 3 h after operations on right paws and 2 days for the right paws. Kruskal–Wallis rank-sum test (non-parametric one-way ANOVA) and Dunn’s test (non-parametric post hoc test) were used. * *p* < 0.05. A total of 6 rats in each of the saline, injection, and tablet groups were used for this analysis.

**Figure 2 pharmaceuticals-15-01586-f002:**
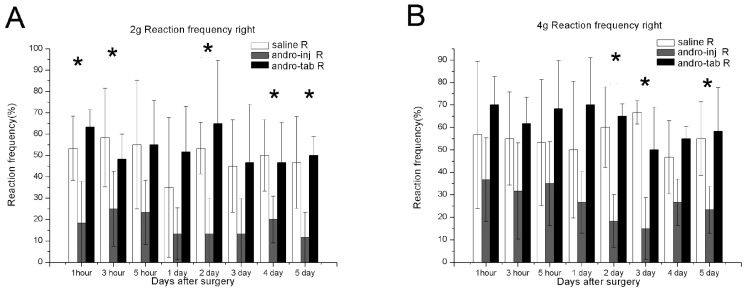
Reaction frequency, an indication of pain sensation, was tested for wounds under different treatment conditions. Mechanical allodynia frequencies were assessed by von Frey test in the right paw (operation site). We chose 2 g hair and 4 g hair to analyze the reaction frequency as these fell within the range that provide the most significant differentiation among treatments. (**A**) When 2 g von Frey hair was used, (**B**) when 4 g von Frey hair was used. For each of the ten tests when using the specific hair, the percentage of withdrawal behavior was noted. The hind paw reaction frequencies were calculated within groups. Saline: rats that had the skin of their right paw cut and sutured, and injected with saline. Andro-inj: rats that had the skin of their right paw cut and sutured, and injected with Andro. Andro-tab: rats that had the skin of their right paw cut, Andro-containing tablets inserted underneath the skin, and sutured. The reaction frequencies of the Andro-inj group were significantly decreased compared to saline group (*), while the Andro-tab group was not significantly different to the saline group, for both 2 g or 4 g hair treated. Kruskal–Wallis rank-sum test (non-parametric one-way ANOVA) and Dunn’s test (non-parametric post hoc test) were used. * *p* < 0.05. A total of 6 rats in each of the saline, injection, and tablet groups were used for this analysis.

**Figure 3 pharmaceuticals-15-01586-f003:**
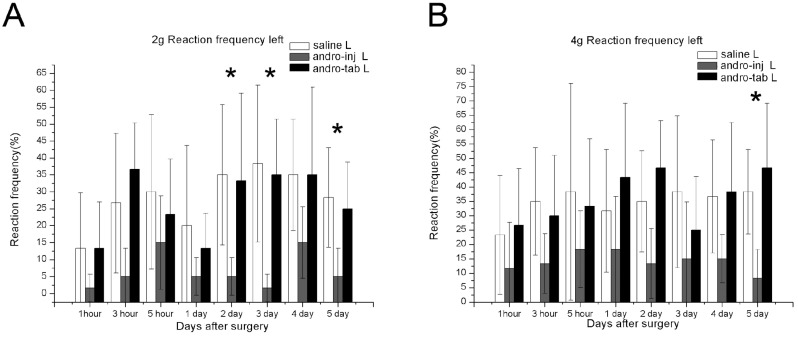
Reaction frequency, an indication of pain sensation, was tested for in wounds under different treatment conditions (left paw). Mechanical allodynia threshold was assessed by von Frey test in the left paw (non-operated feet, for testing mirror pain). We chose 2 g hair and 4 g hair to analyze the reaction frequency as these fell within the range that provided the most significant differentiation among treatments. (**A**) When 2 g von Frey hair was used, (**B**) when 4 g von Frey hair was used. For each of the ten tests when using the specific hair, the percentage of withdrawal behavior exhibited was noted. The hind paw reaction frequencies were averaged within groups. Saline: rats that had the skin of their right paw cut and sutured, and injected with saline. Andro-inj: rats that had the skin of their left paw cut and sutured, and injected with Andro. Andro-tab: rats that had the skin of their left paw cut, Andro-containing tablets inserted underneath the skin, and sutured. In the Andro-inj group, the reaction frequencies were significantly decreased compared to the saline group (*) at several time points, while the Andro-tab group was not significantly different to the saline group. Kruskal–Wallis rank-sum test (non-parametric one-way ANOVA) and Dunn’s test (non-parametric post hoc test) were used. * *p* < 0.05. A total of 6 rats in each of the saline, injection, and tablet groups were used for this analysis.

**Figure 4 pharmaceuticals-15-01586-f004:**
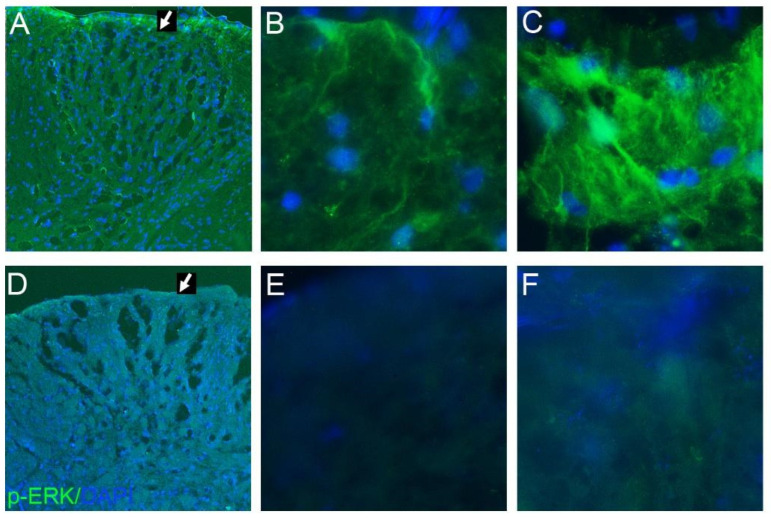
Phospho-ERK staining in the dorsal horn. At the end of 5-days post-surgery, the spinal cord was collected for the study of markers that were regulated in the CNS. The immunostaining of phosphor-ERK1/2 (green) in the laminae I-III in L4-6 areas in saline-treated (**A**–**C**) and Andro-treated (**D**–**F**) rats. (**A**,**D**) Lower magnification photographs displaying the dorsal horn. The photographs shown here are taken from the rats used for von Frey analysis shown in Figure 1, Figure 2 and Figure 3.

**Figure 5 pharmaceuticals-15-01586-f005:**
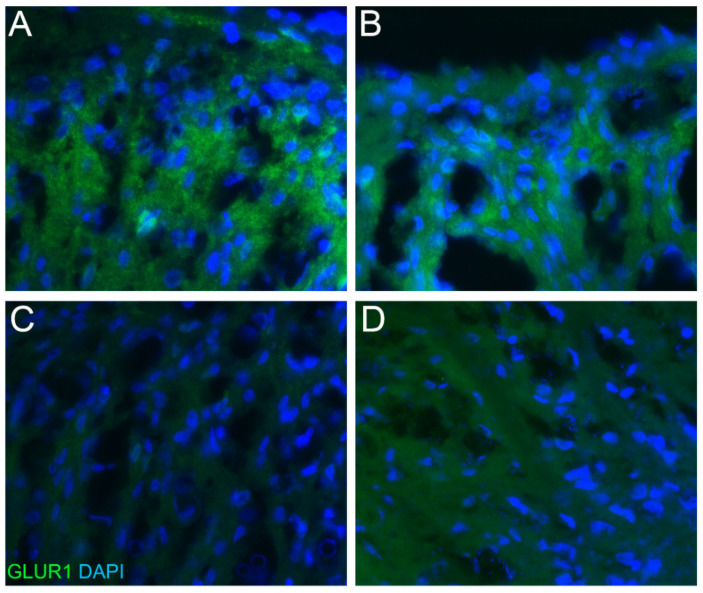
GluR1 staining in the dorsal horn. At the end of 5-days post-surgery, the spinal cord was collected for the study of markers that were regulated in the CNS. The immunostaining of GluR1 (green) in the laminae I–III in L4-6 areas in saline-treated (**A**,**B**) and Andro-treated (**C**,**D**) rats. The staining was most prominent in the neuropil area of the dorsal horn, which contains relatively fewer neurons and some staining was present in the glial cells. The photographs shown here are taken from the rats used for von Frey analysis shown in Figure 1, Figure 2 and Figure 3.

**Figure 6 pharmaceuticals-15-01586-f006:**
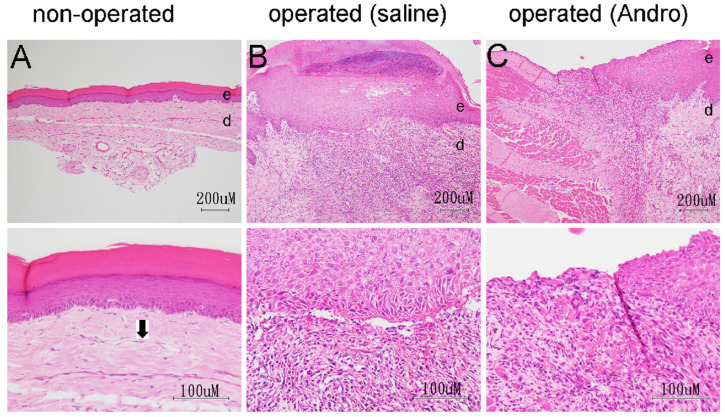
At the end of 5 days post-surgery, the skin was collected from the right paw for the study of the wound healing process. HE staining for representative pictures of the (**A**) non-operated (20L), (**B**) operated, saline-treated (18RS), and (**C**) operated, Andro-injected (21RI). The lower panel is at a higher magnification than the panel above. e: epidermis area. d: dermis area. Arrow in lower panel of A points to a section of the blood vessel. Rats number 20, 18, and 21 were the same rats as those tested for von Frey test.

**Figure 7 pharmaceuticals-15-01586-f007:**
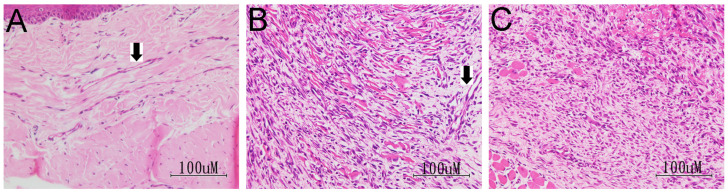
At the end of 5-days post-surgery, the skin was collected from the right paw for the study of the wound healing process. HE staining for representative picture of the (**A**) non-operated (20L), (**B**) operated, Saline-treated (20RS), and (**C**) operated, Andro-injected (21RI). Arrow in (**A**,**B**) points to a section of blood vessel. Rats number 20 and 21 were the same rats as those tested for von Frey test.

**Figure 8 pharmaceuticals-15-01586-f008:**
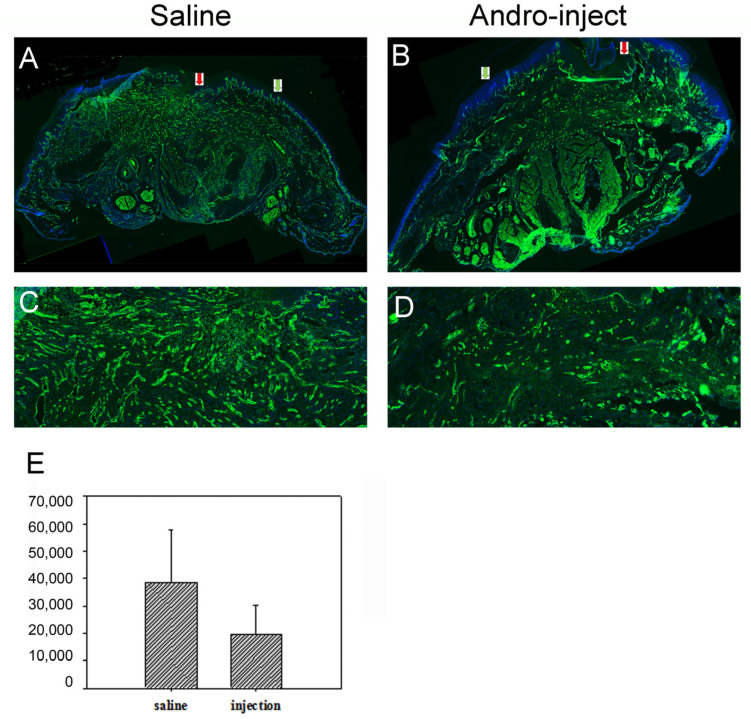
At the end of 5-days post-surgery, the skin was collected from the right paw for the study of the wound healing process and inflammatory reaction. Laminin staining for representative pictures of the (**A**,**C**): saline-treated and (**B**,**D**): Andro-injected groups. (**C**,**D**) high magnification photographs of specific areas from (**A**,**B**). (**E**) in the lower panel shows the statistical analysis of laminin-stained basal lamina from the wound periphery (Wilcoxon rank-sum test, *p* = 0.09958). A total of 6 rats each for the saline and injection groups were used for this analysis. These were the same rats used for the von Frey tests.

**Figure 9 pharmaceuticals-15-01586-f009:**
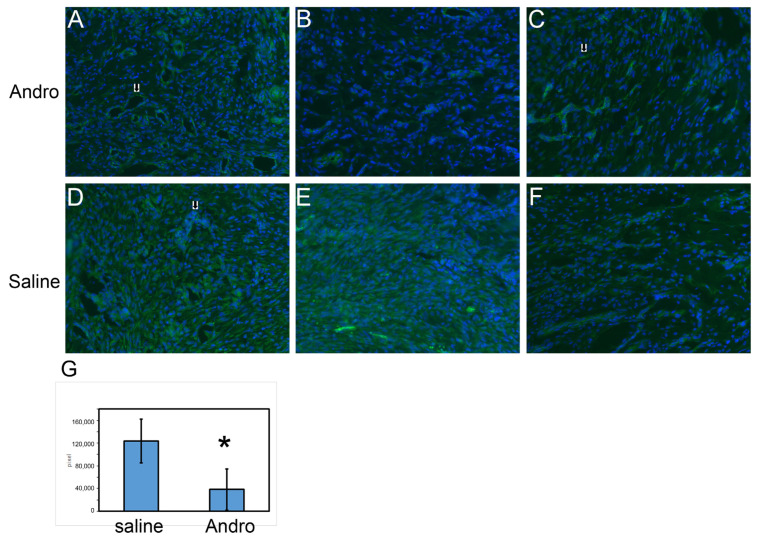
At the end of 5-days post-surgery, the skin was collected from the right paw for the study of the wound healing process. CD31 staining for representative picture of the (**A**–**C**) Andro-injected and (**D**–**F**) saline-injected rats. G in the lower panel shows the statistical analysis conducted with Wilcoxon rank-sum test of CD31-stained blood vessels from the wound periphery. *: *p* < 0.05. A total of 6 rats each for the saline and injection groups were used for this analysis. They are the same rats used for the von Frey tests.

**Figure 10 pharmaceuticals-15-01586-f010:**
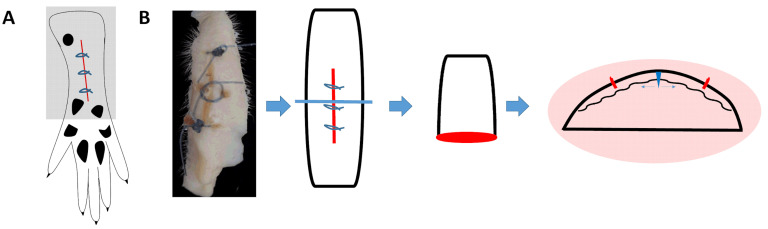
The paw was operated on and sutured as described in Material and methods. The red circle in (**A**) denotes the area where the skin was collected. (**B**), from left to right, a picture of a representative skin sample collected, and a drawing demonstrating the area collected and the plane of the section was marked as red line, followed by the plane of section and a knife to demonstrate plane of section. The image pictured far-right demonstrates again the plane of section. Blue rectangle denotes the area of incision, perpendicular blue arrow denotes the direction of incision and horizontal blue arrow denote the direction where the skin separated from the underlying tissue. Red lines denotes the suture site.

**Figure 11 pharmaceuticals-15-01586-f011:**
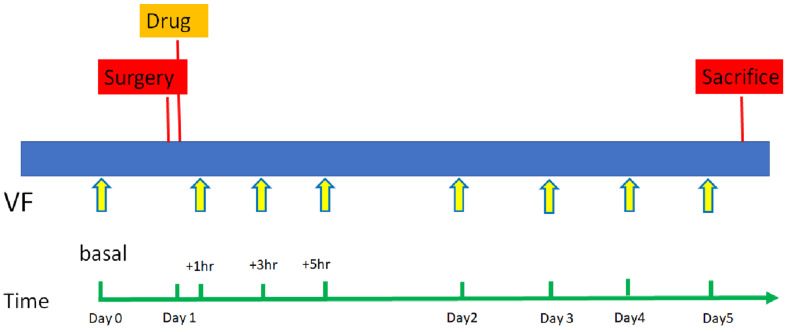
The paw was operated on and sutured as described in Material and methods. Mechanical allodynia threshold at the lateral plantar surface of the hind paw was assessed before wound operation (as basal pain threshold), and then testing began 1 h after surgery and continued at 3, 5, h and 1-, 2-, 3-, 4-, and 5-days post-surgery, and sacrificed at day 5.

**Table 1 pharmaceuticals-15-01586-t001:** Grades for histological findings.

Group	Histopathological Findings	Animal Code											
		Saline											
		L						R					
		9	15	18	20	22	24	9	15	18	20	22	24
Foot pad													
H&E	Crust, epidermis ^1^	0	0	0	0	0	0	0	3	3	2	2	3
	Inflammation, dermis ^1^	0	0	0	0	0	0	4	4	4	4	4	4
	Angiogenesis, dermis ^2^	0	0	0	0	0	0	3	3	3	3	3	3
	Granulation, dermis ^2^	4	4	4	4	4	4	2	2	2	2	2	2
	Re-epithelialization, epidermis ^2^	3	3	3	3	3	3	3	3	3	2	2	2
MT	Granulation, dermis ^2^	4	4	4	4	4	4	2	2	2	2	2	2
Group	Histopathological Findings	Animal Code											
		Injection											
		L						R					
		10	11	12	19	21	23	10	11	12	19	21	23
Foot pad													
H&E	Crust, epidermis ^1^	0	0	0	0	0	0	2	2	3	4	2	3
	Inflammation, dermis ^1^	0	0	0	0	0	0	4	3	4	4	3	4
	Angiogenesis, dermis ^2^	0	0	0	0	0	0	3	3	3	2	2	2
	Granulation, dermis ^2^	4	4	4	4	4	4	2	2	2	2	2	2
	Re-epithelialization, epidermis ^2^	3	3	3	3	3	3	3	3	3	2	2	2
MT	Granulation, dermis ^2^	4	4	4	4	4	4	2	2	2	2	2	2

^1^: Degree of lesions was graded from one to four depending on severity: 1 = minimal (<1%); 2: slight (1–25%); 3 = moderate (26–50%); 4 =moderate/severe (51–75%); 5 = severe/high (76–100%). ^2^: Degree of lesions was graded from one to four according to the method of Altavilla et al., 2001.

**Table 2 pharmaceuticals-15-01586-t002:** Summary of pathological incidence.

Organ	Histopathology	Group	
		Saline	Injection
Foot pad			
	Crust, epidermis ^1^	2.2 ± 1.1	2.7 ± 0.7
	Inflammation, dermis ^1^	4.0 ± 0.0	3.7 ± 0.5
	Angiogenesis, dermis ^1^	3.0 ± 0.0	2.5 ± 0.5 *
	Granulation, dermis ^1^	2.0 ± 0.0	2.0 ± 0.0
	Re-epithelialization, epidermis ^1^	2.5 ± 0.5	2.5 ± 0.5

^1^: Degree of lesions was graded from one to four depending on severity: 1 = minimal (<%); 2: slight (1–25%); 3 = moderate (26–50%); 4 = moderate/severe (51–75%); 5 = severe/high (76–100%). * Statistically significant difference between Saline- and Injection-treated groups set at *p* < 0.05.

## Data Availability

The data presented in this study are available on request from the corresponding author. The data are not publicly available before the publication of this paper.

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
