# Peer review of "Andrographolide Relieves Post-Operative Wound Pain but Affects Local Angiogenesis"

_pharmaceuticals, 2022, doi:10.3390/ph15121586_

Round 1

Reviewer 1 Report

First of all, a significant concern is how the data was collected immediately after a few hours of surgery. The cites for allodynia measurements are also questionable. These are very close to the wounds. 

Secondly, the authors should use an infographic figure to explain their study design and treatment. It is complex and will not be easy for readers. 

Next, the conclusion and discussion are not adequate. They should expand these sections. 

Authors should mention whether they have any animal ethics number and cite it in the manuscript and if their experiments align with ARRIVE guidelines. How they handled their animals and conducted their experiments also needs better clarification.

There are many typographical errors, and the manuscript seems incomplete. Authors should pay attention to the formating and minimise the errors.

Finally, the number of references in the discussion is poor. They should cite relevant papers and discuss similar or conflicting studies. 

Reviewer 2 Report

The manuscript discussed the effect of Andrographolide (Andro) on the reduction of pain sensation in a rat post-operative wound model . It was observed that Andro significantly reduced mechanical allodynia compared to saline-treatment, both in shorter and longer time frame. It influenced expression pERK and GluR1 in the dorsal horn, and the angiogenesis process in the wound healing area.

I accept the manuscript as its but after small changes.

Abstract: Line 20 the plant name Andrographis paniculat should be italic

Line 158 valuesincreased, distance should be added.

Round 2

Reviewer 1 Report

Authors have revised the manuscript and addressed comments.

Author Response

reviewer's comments: 

Authors have revised the manuscript and addressed comments.

My reply:

thank you very much for your comments. I have added a few alterations according to the academic editors comments to upgrad the figure legends.